# Analysis of sensorineural hearing loss in patients attending an otolaryngology clinic in North Central Nigeria

**Nuhu D. Ma'an[1], Ishaku Turaki[1], David Shwe[2], Bulus Nansak[1], Benjamin Babson[1], Simji Gomerep[3], Lauren Malaya[4], David Moffatt[4], Nasim Shakibai[5], Slobodan Paessler[6], Tomoko Makishima[5], Nathan Y. Shehu[3]***

1 Otolaryngology Head and Neck Surgery Department of the Jos University Teaching Hospital, Plateau State, Nigeria, 2 Pediatrics Department of the Jos University Teaching Hospital, Plateau State, Nigeria, 3 Medicine Department of the Jos University Teaching Hospital, Plateau State, Nigeria, 4 School of Medicine, University of Texas Medical Branch, Galveston, Texas, United States of America, 5 Department of Otolaryngology Head and Neck Surgery and Institute for Human Infections and Immunity, University of Texas Medical Branch, Galveston, Texas, United States of America, 6 Department of Pathology and Institute for Human Infections and Immunity, University of Texas Medical Branch, Galveston, Texas, United States of America

* nyshehu@yahoo.com

**Data Availability Statement:** All relevant data are within the paper and its Supporting Information files.

## Abstract

Hearing loss is the third leading cause of years lived with disability. Approximately 1.4 billion people have hearing loss, of which 80% reside in low- and middle-income countries with limited audiology and otolaryngology care available to them. The objective of this study was to estimate period prevalence of hearing loss and audiogram patterns of patients attending an otolaryngology clinic in North Central Nigeria. A 10-year retrospective cohort study was carried out analyzing 1507 patient records of pure tone audiograms of patients at the otolaryngology clinic at Jos University Teaching Hospital, Plateau State, Nigeria. Prevalence of hearing loss of moderate or higher grade increased significantly and steadily after age 60. Compared to other studies, there was a higher prevalence of overall sensorineural hearing loss (24–28% in our study compared to 1.7–8.4% globally) and higher proportions of the flat audiogram configuration among the younger age patients (40% in younger patients compared to 20% in patients older than 60 years). The higher prevalence of the flat audiogram configuration compared to other parts of the world may be suggestive of an etiology specific to this region, such as the endemic Lassa Fever and Lassa virus infection in addition to cytomegalovirus or other viral infections associated with hearing loss.

## Introduction

Hearing loss is currently the third leading cause of years lived with disability [1]. Approximately 1.5 billion people, or 18.7% of the world's population experience hearing loss [1]. About 80% of affected individuals are residing in low-income and middle-income countries, where access to specialty care is often limited [2, 3]. Availability of hearing care services in

**Funding:** This work was partially supported by the National Institutes of Health R01 AI129198 to SP. The funder had no role in study design, data collection and analysis, decision to publish, or preparation of the manuscript.

**Competing interests:** The authors have declared that no competing interests exist.

Africa is scarce compared to other areas of the world. The World Report on Hearing (WRH) estimates that 56% of countries in Africa have fewer than one otolaryngologist per million people and an even lower availability of audiologists with fewer than 1 per million [1, 4]. All African countries with the exception of South Africa, have less than one hearing specialist (otolaryngologist, audiologist, or speech therapist) per 100,000 population [5]. Nigeria has seen increases in the prevalence of otolaryngologists and audiologists from 2009 to 2015, but these professionals are still 0.76 to 0.07 per 100,000 population respectively [5]. Given the rarity of health care providers trained to address hearing loss, it is speculated that Nigerians suffer greater rates of disability from their hearing loss than areas where hearing healthcare services are more accessible and affordable. This constitutes a major public health concern in the largest African country [6].

In West African countries including Nigeria, Lassa virus infection and Lassa fever is one of the major endemic diseases that are known to be associated with severe sensorineural hearing loss (SHL) [6]. While the mortality rate is around 1% and most of those infected will only have minor symptoms, up to 25–30% of those infected were reported to have SHL making Lassa fever the leading infection in the world that causes hearing loss [7].

The overall goal of this study was to determine the period prevalence of and patterns of audiograms of patients attending an otolaryngology clinic in North Central Nigeria, which is known to be endemic for viral infectious diseases that are recognized etiologic agents of SHL.

## Materials and methods

### Ethics statement

Ethical approval for the study was obtained from the Jos University Teaching Hospital ethics committee number JUTH/DCS/IREC/127/XXXI/2549. De-identified age, gender, and audiometry data were used for this study. The study was exempt from obtaining formal consent.

### Subjects

This 10-year retrospective cohort study was carried out between October 2010 and September 2020 at the Otolaryngology Head and Neck Surgery Department of the Jos University Teaching Hospital, Plateau State, Nigeria. The Jos University Teaching Hospital is a tertiary level health center located in North Central Nigeria. It receives referrals from neighboring states of Nasarawa, Bauchi, Benue, Taraba, parts of Kaduna and Kano, with a combined population of over 20 million [8]. The time frame of 10 years chosen for this study as well as the setting of a tertiary hospital was intended to capture as large a portion of the population as possible in order for the findings to have strong external validity and fidelity to the general population in North Central Nigeria.

The inclusion criteria were all patients who visited the otolaryngology clinic with a chief complaint of hearing loss and also had a pure tone audiogram. The exclusion criteria were patients who had a conductive hearing loss of 15dB or greater or had missing data on audiogram, age or gender. The hearing threshold data of 2220 patients were de-identified and then extracted by a trained research assistant and staff member. The audiometry was done using PATH MEDICAL-Sentiero//ADVANCED; MODEL SOH100360. After exclusion of incomplete audiometric data and patients with a 15dB or larger air-bone gap representing conductive hearing loss, data from 1507 patients were available for analysis. Pure tone audiogram air-conduction thresholds (dB) for the left and right ears at each frequency for 250, 500, 1000, 2000, 3000, 4000, and 8000 Hz were used for the analyses. De-identified age, gender, and audiometry data were entered into an Excel spreadsheet (S1 Data) and then exported to Licensed IBM SPSS Inc. Delaware, Chicago, IL, USA, Version 23.0. 1989, 2015, for processing and analysis.

The primary aims of the study were to determine the 1) prevalence of each grade of hearing loss defined by the pure tone average (PTA) as defined in the World Health Organization (WHO) classification and to determine 2) the prevalence of different audiogram patterns in this population. Audiogram patterns are associated with certain etiologies [9].

## Grade of hearing loss analysis

Pure tone average (PTA) was calculated by averaging the thresholds at 500, 1000, 2000, and 4000 Hz for the better hearing ear. This calculation is identical to the method used in the WRH, and was selected to allow comparison with the prevalence reported therein [1]. PTA average was used to assign grades of SHL as normal (<20dB), mild (20 to <35dB), moderate (35 to <50dB), moderately severe (50 to <65dB), severe (65 to <80dB), profound (80 to <95dB), complete SHL or deafness (95dB or greater), or unilateral SHL (<20dB in better ear and 35dB or greater in worse ear) based on WHO classification [10]. The WHO grading system of hearing loss is a validated scoring tool for age-related hearing loss that predicts deficits in communication performance in unaided patients [11].

The percentage of patients with each grade of hearing loss was calculated and categorized into the following age groups: 8–19 years, 20–29 years, 30–39 years, 40–49 years, 50–59 years, 60–69 years, 70–79 years, 80–90 years, and 90+ years. Patients were categorized by gender within each age group and the proportions based on age group and gender were calculated. The number of patients with high grade hearing loss (the sum of patients in each age group with moderate, moderately severe, severe, profound, and complete hearing loss) was calculated. High grade hearing loss patients were further grouped by gender and by age group.

## Audiogram pattern

Frequencies 250, 500, 1000, 2000, 3000, 4000, and 8000 Hz for the left and right ears were used to reproduce audiograms with a custom MATLAB (version: 2019b) script. The audiograms were then categorized into six patterns. The pattern of the audiogram was determined by assessing the shape made by a line drawn between thresholds at each frequency. Data from the better hearing ear were used to assign the pattern of the audiogram.

The six patterns used for categorization of the better ear were "flat", "downward sloping", "upward sloping", "tent-shaped", "cookie bite", and "unclassified". A flat pattern was characterized by a difference within 20 dB between the highest and lowest threshold across all frequencies [9]. Audiograms with one outlier in the 250 Hz or 8000 Hz frequencies that fell outside 20 dB difference but were otherwise within this margin were categorized as flat. Downward sloping patterns had lower minimum hearing thresholds in the low frequency range and had increasing thresholds as the frequency increased for at least 3 consecutive frequencies with a difference of greater than 20 dB across all frequencies. Upward sloping patterns had lower minimum hearing thresholds in the higher frequencies with increasing thresholds as the frequency decreased with an overall difference of greater than 20 dB across 3 or more frequencies. Cookie-bite patterns had the lowest thresholds at both low and high frequencies with an increased threshold required to detect tones in the middle ranges (1000, 2000, and 3000 Hz). Again, this difference between the better-detected ends of the range and the middle range had to be greater than 20 dB. Tent-shaped patterns had the lowest threshold in the middle of the range and the highest threshold at the low and high frequencies with a greater than 20 dB difference between the ends and middle frequencies. Audiograms were categorized as unclassified if the data did not resemble a pattern listed previously.

## Statistical analysis

Statistical analysis was conducted using SPSS Statistics. The percentages of each audiogram pattern and grade of hearing were separated by sex and age group. Association between these variables was analyzed using a chi-square test which revealed a statistically significant difference ($X^2$ = 136.25, p<0.05, 2-sided).

## Results

Out of 1507 patients analyzed, the male to female ratio was 1.0:1.1. The age distribution of these patients is summarized in Table 1. Between the ages of 8–49 years old, the majority of the patients had hearing within normal limits with over half (52%) in the 20–29 year-old group. From 50–69 years old, mild hearing loss was the most prevalent grade of hearing loss, most notably making up 38% in 50–59 year-old patients. In the older age groups, moderate and moderately severe grade hearing loss was common: 29% and 23% at 70–79 years, 16% and 53% at 80–89 years, and 33% and 25% at 90+ years of age groups respectively (Fig 1). In age groups 60 and older, the prevalence of patients with moderate or higher grade hearing loss increased significantly, whereas unilateral hearing loss was more frequently seen at ages less than 60 (Table 2). Audiogram patterns showed specific patterns at different age groups. There was significantly higher prevalence of patients with downward sloping patterns in age group 60 and older compared to ages 8–29 years old. On the other hand, there was significantly higher prevalence of upward sloping pattern in patients 8–29 years compared to age 60 and older. The cookie bite pattern and flat pattern was less prevalent in age group 60 and older (Table 3).

Prevalence of significant hearing loss of moderate or higher grade in ages 8 to 59 years in this population was relatively stable at around 24–28%. At age 60, the percentage of people with moderate or greater hearing loss increased steadily up to 91% at ages 80–89 years (Fig 2). Both males and females showed a similar trend, but hearing loss was more prevalent in males in most age groups.

## Audiogram pattern

When categorized by age, flat audiograms were the most prevalent pattern in younger age groups: 41% in ages 8–30, and 40% in ages 31–60. In the 61+ age group, downward sloping pattern was the most prevalent audiogram pattern at 49% (Fig 3). Across all age groups, the other patterns were relatively constant: tent-shaped (13% in age 8–30, 15% in age 31–60, 10% in age 61+), cookie-bite (7% in age 8–30, 4% in age 31–60, 2% in age 61+), and unclassified

**Table 1. Demographics of patients in the analysis.**

| Age | Number of patients (Males, Females) | Percentage of Sample |
|---|---|---|
| 8–19 | 180 (83, 97) | 11.9 |
| 20–29 | 258 (126, 132) | 17.1 |
| 30–39 | 263 (124, 139) | 17.5 |
| 40–49 | 248 (127, 121) | 16.5 |
| 50–59 | 252 (96, 156) | 16.7 |
| 60–69 | 163 (86, 77) | 10.8 |
| 70–79 | 99 (56, 43) | 6.6 |
| 80–89 | 32 (21, 11) | 2.1 |
| 90+ | 12 (7, 5) | 0.8 |
| *Total* | **1507 (726, 781)** | **100** |

**Fig 1. Grade of hearing loss by age.** PTA (sum of males and females) was used to determine the prevalence of hearing loss in different age groups. The percentage of patients whose hearing is within normal limits decreased with age, while the percentage of higher grades of hearing loss was more prevalent in older ages.

(9% in age 8–30, 7% in age 31–60, 13% for age 61+). The upward sloping pattern showed decline as the age of patients increased (11% in age 8–30, 7% in ages 31–60, 1% in age 61+). The trend was similar between males and females.

## Discussion

This is the largest population-based study of hearing loss prevalence in a Lassa fever endemic region in Central Nigeria. This study demonstrated that hearing loss increases as early as age 40 in this population, as seen in the increase in mild hearing loss in the patients in their 40s (Fig 1). These findings are consistent with studies assessing hearing loss in other countries [12–16] with one report suggesting an even earlier onset of age-related hearing loss [17]. The majority of patients with high grade hearing loss were in age cohorts older than 60 years (Fig 2). Any reduction of hearing capacity with disability should be investigated seriously due to its close association with declines in social function [18]. Combined with the findings in the WHR on hearing [1], our data show roughly half or greater of the elderly population are experiencing SHL significant enough to affect social function. These findings demonstrate the importance of instituting hearing care services to reduce hearing loss-related disability and its associated profound social effects.

The prevalence of moderate or higher levels of hearing loss, on a global scale, and as reported by WHO [1], are slightly higher among males than females with 217 million males (5.6%) and 211 million females (5.5%) with hearing loss [10]. The WHO reported that nearly two-thirds of females began to show signs of mild hearing loss earlier, starting in their 50s, while the majority of males did not show this trend until later around age 60. This observation has been corroborated in previous studies from other parts of the world. The reason for this observation is unclear but this might be caused by more efficient health-seeking behavior in females compared to male counterparts [19, 20]. Therefore, gender differences should be taken into consideration when instituting intervention programs, such as earlier screening for hearing loss in females or encouraging males to undergo regular hearing screenings.

**Table 2. Summary of grade of hearing loss by age.**

| | Age (years) | | | | | | | | | |
| | 8–19 | 20–29 | 30–39 | 40–49 | 50–59 | 60–69 | 70–79 | 80–89 | 90+ | |
|---|---|---|---|---|---|---|---|---|---|---|
| **Grade** | | | | | | | | | | Total |
| **Normal** | n = 49 | n = 76 | n = 86 | n = 70 | n = 55 | n = 20 | n = 4 | n = 0 | n = 1 | n = 361 |
| % in Grade | 13.6% | 21.1% | 23.8% | 19.4% | 15.2% | 5.5% | 1.1% | 0.0% | 0.3% | 100% |
| % in Age | 27.2% | 29.5% | 32.7% | 28.2% | 21.8% | 12.3% | 4.0% | 0.0% | 8.3% | 24.0% |
| **Mild** | n = 28 | n = 37 | n = 37 | n = 65 | n = 96 | n = 57 | n = 21 | n = 3 | n = 2 | n = 346 |
| % in Grade | 8.1% | 10.7% | 10.7% | 18.8% | 27.7% | 16.5% | 6.1% | 0.9% | 0.6% | 100% |
| % in Age | 15.6% | 14.3% | 14.1% | 26.2% | 38.1% | 35.0% | 21.2% | 9.4% | 16.7% | 23.0% |
| **Moderate** | n = 12 | n = 19 | n = 19 | n = 27 | n = 28 | n = 28 | n = 29 | n = 5 | n = 4 | n = 171 |
| % in Grade | 7.0% | 11.1% | 11.1% | 15.8% | 16.4% | 16.4% | 17.0% | 2.9% | 2.3% | 100% |
| % in Age | 6.7% | 7.4% | 7.2% | 10.9% | 11.1% | 17.2% | 29.3% | 15.6% | 33.3% | 11.3% |
| **Moderately severe** | n = 19 | n = 21 | n = 25 | n = 8 | n = 20 | n = 27 | n = 23 | n = 17 | n = 3 | n = 163 |
| % in Grade | 11.7% | 12.9% | 15.3% | 4.9% | 12.3% | 16.6% | 14.1% | 10.4% | 1.8% | 100% |
| % in Age | 10.6% | 8.1% | 9.5% | 3.2% | 7.9% | 16.6% | 23.2% | 53.1% | 25.0% | 10.8% |
| **Severe** | n = 11 | n = 12 | n = 13 | n = 12 | n = 5 | n = 9 | n = 13 | n = 3 | n = 1 | n = 79 |
| % in Grade | 13.9% | 15.2% | 16.5% | 15.2% | 6.3% | 11.4% | 16.5% | 3.8% | 1.3% | 100% |
| % in Age | 6.1% | 4.7% | 4.9% | 4.8% | 2.0% | 5.5% | 13.1% | 9.4% | 8.3% | 5.2% |
| **Profound** | n = 3 | n = 7 | n = 4 | n = 2 | n = 2 | n = 3 | n = 1 | n = 3 | n = 1 | n = 26 |
| % in Grade | 11.5% | 26.9% | 15.4% | 7.7% | 7.7% | 11.5% | 3.8% | 11.5% | 3.8% | 100% |
| % in Age | 1.7% | 2.7% | 1.5% | 0.8% | 0.8% | 1.8% | 1.0% | 9.4% | 8.3% | 1.7% |
| **Complete** | n = 6 | n = 10 | n = 13 | n = 11 | n = 6 | n = 6 | n = 3 | n = 1 | n = 0 | n = 56 |
| % in Grade | 10.7% | 17.9% | 23.2% | 19.6% | 10.7% | 10.7% | 5.4% | 1.8% | 0.0% | 100% |
| % in Age | 3.3% | 3.9% | 4.9% | 4.4% | 2.4% | 3.7% | 3.0% | 3.1% | 0.0% | 3.7% |
| **Unilateral** | n = 52 | n = 76 | n = 66 | n = 53 | n = 40 | n = 13 | n = 5 | n = 0 | n = 0 | n = 305 |
| % in Grade | 17.0% | 24.9% | 21.6% | 17.4% | 13.1% | 4.3% | 1.6% | 0.0% | 0.0% | 100%5 |
| % in Age | 28.9% | 29.5% | 25.1% | 21.4% | 15.9% | 8.0% | 5.1% | 0.0% | 0.0% | 20.2% |
| Total | n = 180 | n = 258 | n = 263 | n = 248 | n = 252 | n = 163 | n = 99 | n = 32 | n = 12 | n = 1507 |
| % in Grade | 11.9% | 17.1% | 17.5% | 16.5% | 16.7% | 10.8% | 6.6% | 2.1% | 0.8% | 100% |
| % in Age | 100% | 100% | 100% | 100% | 100% | 100% | 100% | 100% | 100% | 100% |

**Pearson Chi-Square:** Value: 520.208; dF = 56; Asymptomatic Significance (2-sided): P = 0.000
**N of Valid Cases:** 1507

The prevalence of moderate to severe hearing loss among patients aged 8–59 years in our study is comparable to the 25% prevalence reported from the National Ear Care Center, Kaduna, North-West Nigeria, but differed significantly from 1.7–8.4% reported from the WHR on hearing [1, 21]. This finding reveals a high prevalence of hearing loss in North Central Nigeria and highlights the vast need for investment into hearing health at a population level.

We observed an increased prevalence of hearing loss with high prevalence of the downward sloping pattern. This observation is consistent with earlier published studies in the literature [22–24]. Audiogram configuration is a recognized non-invasive diagnostic tool that potentially lends clues to the etiology of hearing deficit among the different types of sensorineural hearing loss. The shape of hearing thresholds plotted across the frequency spectrum is very important for better understanding the type of hearing loss [23]. For example, a downward sloping pattern showing more hearing loss at high frequencies could be indicative of presbycusis [9]. Other patterns such as flat, upward sloping, cookie-bite, and tent shaped configurations are

**Table 3. Summary of audiogram pattern and age.**

| | Age (years) | | | |
|---|---|---|---|---|
| | **8–29** | **30–59** | **60 +** | |
| **Audiogram pattern** | | | | Total |
| **Upward sloping** | n = 49 | n = 52 | n = 3 | n = 104 |
| % in pattern | 47.1% | 50.0% | 2.9% | 100.0% |
| % in age | 11.2% | 6.8% | 1.0% | 6.9% |
| **Downward sloping** | n = 64 | n = 169 | n = 139 | n = 372 |
| % in pattern | 17.2% | 45.4% | 37.4% | 100.0% |
| % in age | 14.6% | 22.1% | 45.4% | 24.7% |
| **Flat** | n = 168 | n = 302 | n = 79 | n = 549 |
| % in pattern | 30.6% | 55.0% | 14.4% | 100.0% |
| % in age | 38.4% | 39.6% | 25.8% | 36.4% |
| **Tent shaped** | n = 60 | n = 105 | n = 26 | n = 191 |
| % in pattern | 31.4% | 55.0% | 13.6% | 100.0% |
| % in age | 13.7% | 13.8% | 8.5% | 12.7% |
| **Cookie bite** | n = 14 | n = 19 | n = 1 | n = 34 |
| % in pattern | 41.2% | 55.9% | 2.9% | 100.0% |
| % in age | 3.2% | 2.5% | 0.3% | 2.3% |
| **Unclassified** | n = 37 | n = 46 | n = 36 | n = 119 |
| % in pattern | 31.1% | 38.7% | 30.3% | 100.0% |
| % in age | 8.4% | 6.0% | 11.8% | 7.9% |
| **Unassigned/Not tested** | n = 46 | n = 70 | n = 22 | n = 138 |
| % in pattern | 33.3% | 50.7% | 15.9% | 100.0% |
| % in Age | 10.5% | 9.2% | 7.2% | 9.2% |
| Total | n = 438 | n = 763 | n = 306 | n = 1507 |
| % in pattern | 29.1% | 50.6% | 20.3% | 100.0% |
| % in age | 100.0% | 100.0% | 100.0% | 100.0% |

**Pearson Chi-Square:** Value: 136.245; dF = 12; Asymptomatic Significance (2-sided): P = 0.000 **N of Valid Cases:** 1507

also commonly utilized to characterize hearing loss patterns [23]. While audiograms can suggest etiologies for hearing loss unique to this population, this has not been studied in the Nigerian population. Therefore, pertaining to our interest in Lassa virus infection etiology, there are no population-based studies that can be used to determine a common audiogram pattern. The lack of other demographic information such as medical and social history results in difficulty associating etiologies with hearing loss especially if patients have multiple possible causative factors. This was a limitation in our study. However, even with the limitation we were able to reveal the higher prevalence of hearing loss.

One characteristic finding in our cohort was the abundance of the flat audiogram configuration. Our cohort had approximately twice the frequency of flat configuration audiograms at 12% (excluding hearing within normal limits) as compared to a study reported in the United States at less than 5% [24]. Such flat audiogram configuration is seen in cytomegalovirus infection-induced hearing loss: the most common viral infection-induced hearing loss worldwide [25], which is characterized by sudden onset SHL and is associated with worse prognostic outcome. If viral infection mediated hearing loss share similar audiogram configurations, our speculation is that an endemic Lassa virus infection can be contributing to the additional number of patients with SHL in this region.

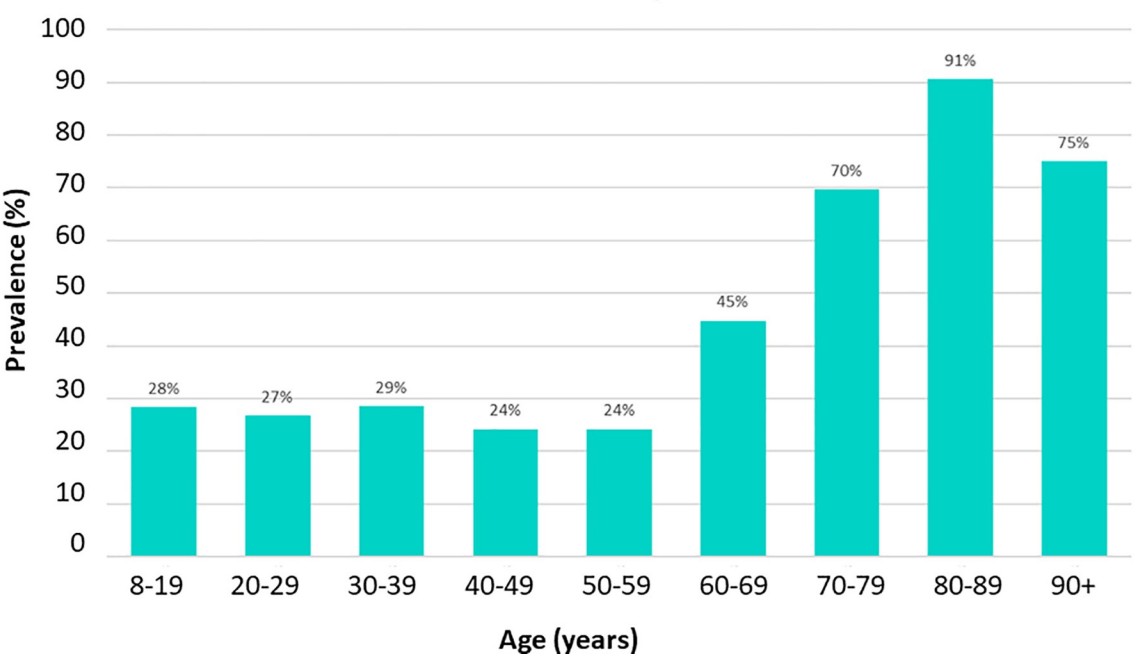

**Fig 2. Prevalence of hearing loss of moderate or higher grade by age.** PTA (sum of males and females) was used to determine the prevalence of hearing loss in different age groups. Greater increases in moderate or severe hearing loss are seen in patients ages 60 and older.

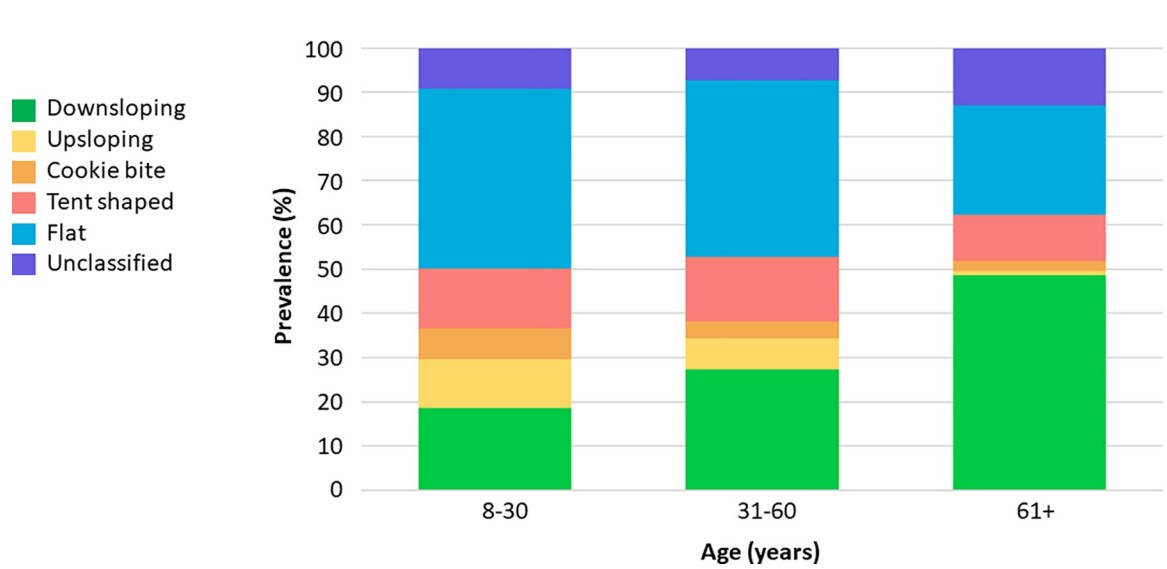

**Fig 3. Audiogram pattern by age.** Downward sloping pattern increases with age while flat pattern and upsloping patterns decrease with age.

Overall, our results show a higher prevalence of hearing loss in North Central Nigeria as compared to other nations. This difference may be attributable to the endemic Lassa virus infection which is known to cause SHL. This primary analysis on the past 10 years of audiological data is not sufficient to draw any conclusions. More information including epidemiological and serological analyses is needed to confirm this hypothesis and to establish a causal link. Further scientific pursuit should be revisited when funding for otology and audiology care services are addressed, which is the greater needs of these patients.

## Conclusions

This study observed a high prevalence of moderate or higher degree sensorineural hearing loss among younger age patients and higher than expected flat audiogram configurations in a region endemic for Lassa virus infections. These differences in prevalence from other published studies in other nations suggest higher rates of non-age-related SHL in North Central Nigeria. More information is required to further investigate the causal link between endemic virus infections and SHL in North Central Nigeria, particularly the burden imposed by endemic Lassa virus.

## Supporting information

**S1 Data. De-identified data on age, gender, and auditory test used for this study.**
(XLSX)

## Author Contributions

**Conceptualization:** Nuhu D. Ma'an, David Shwe, Tomoko Makishima, Nathan Y. Shehu.

**Data curation:** Nuhu D. Ma'an, Ishaku Turaki, David Shwe, Bulus Nansak, Simji Gomerep, Lauren Malaya, David Moffatt.

**Formal analysis:** Nuhu D. Ma'an, Benjamin Babson, Lauren Malaya, David Moffatt, Nasim Shakibai.

**Funding acquisition:** Slobodan Paessler.

**Investigation:** Nuhu D. Ma'an, Ishaku Turaki, Bulus Nansak, Benjamin Babson, Lauren Malaya, David Moffatt, Nasim Shakibai.

**Methodology:** Lauren Malaya, David Moffatt.

**Resources:** Nathan Y. Shehu.

**Software:** Lauren Malaya.

**Supervision:** David Shwe, Simji Gomerep, Slobodan Paessler, Tomoko Makishima, Nathan Y. Shehu.

**Validation:** Nuhu D. Ma'an, Benjamin Babson, David Moffatt, Tomoko Makishima.

**Visualization:** Benjamin Babson, Lauren Malaya, David Moffatt.

**Writing – original draft:** Lauren Malaya, David Moffatt.

**Writing – review & editing:** Nuhu D. Ma'an, Ishaku Turaki, David Shwe, Bulus Nansak, Benjamin Babson, Simji Gomerep, Nasim Shakibai, Slobodan Paessler, Tomoko Makishima, Nathan Y. Shehu.

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
