## [Decision Letter · Decision Letter 0]

17 Jan 2022

PGPH-D-21-00966

A ten-year retrospective review of audiogram configurations of patients attending an otolaryngology clinic in Lassa fever endemic North Central Nigeria.

Dear Dr. Makishima,

Thank you for submitting your manuscript to PLOS Global Public Health. After careful consideration, we feel that it has merit but does not fully meet PLOS Global Public Health’s publication criteria as it currently stands. Therefore, we invite you to submit a revised version of the manuscript that addresses the points raised during the review process.

We look forward to receiving your revised manuscript.

Kind regards,

Nomfundo Moroe

Academic Editor

Journal Requirements:

1. Please ensure that the funders and grant numbers match between the Financial Disclosure field and the Funding Information tab in your submission form. Note that the funders must be provided in the same order in both places as well.

2. Please update your Competing Interests statement. If you have no competing interests to declare, please state: “The authors have declared that no competing interests exist.”

3. Please provide a complete Data Availability Statement in the submission form, ensuring you include all necessary access information or a reason for why you are unable to make your data freely accessible. Note that it is not acceptable for the authors to be the sole named individuals responsible for ensuring data access.

PLOS defines a study's minimal data set as the underlying data used to reach the conclusions drawn in the manuscript and any additional data required to replicate the reported study findings in their entirety. Any potentially identifying patient information must be fully anonymized. 

If your research concerns only data provided within your submission, please write “All data are in the manuscript and/or supporting information files” as your Data Availability Statement.

4. Please remove any figures embedded in your manuscript file.

Additional Editor Comments (if provided):

Reviewers' comments:

Reviewer's Responses to Questions

**Comments to the Author**

1. Does this manuscript meet PLOS Global Public Health’s publication criteria? Is the manuscript technically sound, and do the data support the conclusions? The manuscript must describe methodologically and ethically rigorous research with conclusions that are appropriately drawn based on the data presented.

Reviewer #1: Partly

Reviewer #2: Yes

2. Has the statistical analysis been performed appropriately and rigorously?

Reviewer #1: No

Reviewer #2: Yes

3. Have the authors made all data underlying the findings in their manuscript fully available (please refer to the Data Availability Statement at the start of the manuscript PDF file)?

Reviewer #1: Yes

Reviewer #2: No

4. Is the manuscript presented in an intelligible fashion and written in standard English?

Reviewer #1: No

Reviewer #2: No

5. Review Comments to the Author

Reviewer #1: Reviewer’s comments:

Journal: PLOS Global Public Health

Manuscript number: PGPH-D-21-00966

Manuscript title: A ten-year retrospective review of audiogram configurations of patients attending an otolaryngology clinic in Lassa fever endemic North Central Nigeria.

Thank you for the opportunity to review this interesting manuscript which promises to yield useful information and offers insight into a necessary area of research.

INTRODUCTION AND LITERATURE REVIEW

• This section offers insight into the context and sets the scene for the study.

METHODOLOGY

• The authors state, “Specifically, the study assessed the burden of moderate or higher degree hearing loss at different age groups…” but the method does not delineate how burden is measured since it is not only an audiometric concept. Moreover, the aims could be specifically detailed.

• The time frame of the review requires substantiation, justification, and expansion while accounting for its contemporaneity.

• The data collection describes the assessment of patients, but this manuscript is described as a review so the methodology of the review of the patient data ought to be described i.e. how data were accessed, selected, cleaned, mined, recorded, permissions, by whom, etc. etc.

• Inclusion and exclusion criteria need to be elucidated.

• Consideration needs to be given to the reliability of the PTA as “PTA average was used to assign grades of hearing loss” (page 8) especially since there is no mention of what was considered in cases of big differences between adjacent frequencies, for example.

• Please offer a reference for the configurations of the audiograms.

• It is stated that analysis was conducted on SPSS but the specific analyses are not elucidated, especially with related to descriptive and inferential statistics, especially since in the results, the term ‘trend’ is used as is ‘disproportionately’ while there’s also reference to ‘twice the frequency’ but little detail is provided of that significance.

RESULTS, DISCUSSION & CONCLUSION

• Reference is made to normal hearing but, although hearing may be within normal limits, there may still be hearing or listening difficulties as acknowledged in the discussion when it is stated, “Any reduction of hearing capacity with disability should be investigated seriously due to its close association with declines in social function.” so please look at replacing ‘normal hearing’ with ‘hearing within normal limits’ as delineated by the analyses.

• The intention is stated as a measurement of burden but the results are reflective of audiometric configurations, degree, etc. but the burden, especially since reference was made to “years lived with disability.”

• The comments related to the methodology feed into the review of the results and reflect the same need for further exegesis.

• The conclusion ought to be strengthened in relation to the aforementioned comments while also delineating conclusions which strongly emanate from the findings and the intention of this project.

FORMAT and STYLE

• Please replace the term ‘subjects’ to refer to the patient records, or records, etc. and patients instead of subjects when referring to people.

• Please replace ‘down sloping’ with ‘downward sloping’ and ‘upsloping’ to ‘upward sloping’

• Please spell out acronyms in full the first time before using the acronym.

• Please use fewer instead of less where appropriate for items which can be counted e.g. “… and an even lower availability of audiologists with less than 1 per million…” amongst others.

• Please use consistent terminology e.g. ENT and otolaryngologist if it applies (since technically otolaryngologist is not synonymous with ENT but otorhinolaryngologist would b a closer match).

• Please use lower case letters for common nouns e.g. an Otolaryngology Clinic

• Please check for typographical errors e.g. Sentiero//ADVACED ought to be ADVANCED.

• The word ‘data’ is a plural noun so needs to be accompanied by a plural verb.

OVERALL EVALUATION

• This paper could have potential value. However, in its present form, although interesting, requires amendment for further consideration as potentially suitable for publication in the journal. The recommendation is MAJOR REVISION.

Reviewer #2: Thank you for opportunity to review this submission. This interesting article highlights the higher than average prevalence of hearing loss in this area of Nigeria. I've made comments on the submission, but would like to highlight the following:

- The title of the manuscript is misleading as no information was presented regarding the medical history (specifically related to Lassa Fever). The manuscript only addressed sensorineural hearing loss, thus discarding the information on conductive hearing loss. This (SNHL) should be reflected in the title

- Significant editorial care (typographical and grammatical) is required. The consistent use of terminology is encouraged (degree/grade; ENT/otolaryngologist, etc.). Although standard English is used, it is sometimes difficult to understand - have the manuscript professional edited before the resubmission.

- If the authors decide to change the title, the introduction section should also be amended. A suggestion is to start with information presented from lines 54- 67 and then highlight the potential contributing factors to hearing loss such as Lassa Fever, CMV, presbycusis, etc. in North Central Nigeria.

- Methodology: The lack of more demographic information (e.g., medical history on potential cause of hearing loss) is a limitation of this study. Further, only descriptive statistics were used, would the use on inferential statistics not strengthen the findings?

- The discussion of the findings can be more robust and contextually relevant and should be amended based on the decision made by the authors regarding the title, etc.

6. PLOS authors have the option to publish the peer review history of their article (what does this mean?). If published, this will include your full peer review and any attached files.

**Do you want your identity to be public for this peer review?** For information about this choice, including consent withdrawal, please see our Privacy Policy.

Reviewer #1: No

Reviewer #2: No

---

## [Decision Letter · Decision Letter 1]

19 Jan 2023

PGPH-D-21-00966R1

Analysis of sensorineural hearing loss in patients attending an otolaryngology clinic in North Central Nigeria.

Dear Dr. Makishima,

Thank you for submitting your manuscript to PLOS Global Public Health. After careful consideration, we feel that it has merit but does not fully meet PLOS Global Public Health’s publication criteria as it currently stands. Therefore, we invite you to submit a revised version of the manuscript that addresses the points raised during the review process.

Your manuscript has been reassessed by one reviewer from the previous round, whose comments can be found in the attached pdf file. Please ensure you respond to each point carefully in your response to reviewers, and modify your manuscript accordingly.

We look forward to receiving your revised manuscript.

Kind regards,

Dr Joseph Donlan

Editorial Office

Journal Requirements:

Additional Editor Comments (if provided):

Reviewers' comments:

Reviewer's Responses to Questions

**Comments to the Author**

1. If the authors have adequately addressed your comments raised in a previous round of review and you feel that this manuscript is now acceptable for publication, you may indicate that here to bypass the “Comments to the Author” section, enter your conflict of interest statement in the “Confidential to Editor” section, and submit your "Accept" recommendation.

Reviewer #2: All comments have been addressed

2. Does this manuscript meet PLOS Global Public Health’s publication criteria? Is the manuscript technically sound, and do the data support the conclusions? The manuscript must describe methodologically and ethically rigorous research with conclusions that are appropriately drawn based on the data presented.

Reviewer #2: Yes

3. Has the statistical analysis been performed appropriately and rigorously?

Reviewer #2: No

4. Have the authors made all data underlying the findings in their manuscript fully available (please refer to the Data Availability Statement at the start of the manuscript PDF file)?

Reviewer #2: Yes

5. Is the manuscript presented in an intelligible fashion and written in standard English?

Reviewer #2: Yes

6. Review Comments to the Author

Reviewer #2: Attachment included with comments

7. PLOS authors have the option to publish the peer review history of their article (what does this mean?). If published, this will include your full peer review and any attached files.

**Do you want your identity to be public for this peer review?** For information about this choice, including consent withdrawal, please see our Privacy Policy.

Reviewer #2: No

---

## [Editor Report · Decision Letter 2]

9 Feb 2023

Analysis of sensorineural hearing loss in patients attending an otolaryngology clinic in North Central Nigeria.

PGPH-D-21-00966R2

Dear Dr Makishima,

We are pleased to inform you that your manuscript 'Analysis of sensorineural hearing loss in patients attending an otolaryngology clinic in North Central Nigeria.' has been provisionally accepted for publication in PLOS Global Public Health.

Best regards,

Julia Robinson

Executive Editor